**Investigation**

# Substitution load revisited: a high proportion of deaths can be selective

Joseph Matheson [ID] [1,2] Moises Exposito-Alonso [ID] [3,4] Joanna Masel [ID] [1,*]

[1]Department of Ecology and Evolutionary Biology, University of Arizona, Tucson, AZ 85721, USA
[2]Department of Ecology, Behavior, and Evolution, University of California San Diego, San Diego, CA 92093, USA
[3]Departments of Plant Biology & Global Ecology, Carnegie Institution for Science, Stanford University, Stanford, CA 94305, USA
[4]Present address: Department of Integrative Biology, University of California Berkeley, Berkeley, CA 94720, USA

*Corresponding author: Department of Ecology and Evolutionary Biology, University of Arizona, 1041 E Lowell St, Tucson, AZ 85721, USA. Email: masel@arizona.edu

Haldane's Dilemma refers to the concern that the need for many "selective deaths" to complete a substitution (i.e. selective sweep) creates a speed limit to adaptation. However, discussion of this concern has been marked by confusion, especially with respect to the term "substitution load". Here, we distinguish different historical lines of reasoning, and identify one, focused on finite reproductive excess and the proportion of deaths that are "selective" (i.e. causally contribute to adaptive allele frequency changes), that has not yet been fully addressed. We develop this into a more general theoretical model that can apply to populations with any life history, even those for which a generation or even an individual are not well defined. The actual speed of adaptive evolution is coupled to the proportion of deaths that are selective. The degree to which reproductive excess enables a high proportion of selective deaths depends on the details of when selection takes place relative to density regulation, and there is therefore no general expression for a speed limit. To make these concepts concrete, we estimate both reproductive excess, and the proportion of deaths that are selective, from a dataset measuring survival of 517 different genotypes of *Arabidopsis thaliana* grown in 8 different environmental conditions. In this dataset, a much higher proportion of deaths contribute to adaptation, in all environmental conditions, than the 10% cap that was anticipated as substantially restricting adaptation during historical discussions of speed limits.

Keywords: cost of selection; adaptation rate; genetic load; fitness component; biological individual

## Introduction

During an adaptive sweep, new alleles must be substituted for old alleles across an entire population. This means that individuals with the old alleles must leave no descendants that carry them, and individuals with new alleles must produce enough offspring to replenish the population. These requirements limit the number of sweeps that can happen within a given time. Haldane (1957) used this reasoning to propose a rough estimate of the maximum speed of adaptation.

Haldane's arguments were the historical motivation for the development of Kimura's (Kimura 1968) neutral theory. This speed limit became known as Haldane's dilemma (Van Valen 1963) because data on rates of amino acid divergence between species seemed to exceed Haldane's speed limit. The development of neutral theory resolved this apparent dilemma by suggesting that most amino acid substitutions are neutral and do not count against the speed limit. However, the basis for this historical argument is now on troubled ground, because recent literature argues that the fraction of amino acid substitutions explained by adaptation can be high (Sella *et al.* 2009; Galtier 2016; Uricchio *et al.* 2019; Murga-Moreno *et al.* 2024). On shorter timescales, recent experiments have shown rapid seasonal adaptation in *Drosophila* (Bertram 2021; Machado *et al.* 2021; Kelly 2022; Bitter *et al.* 2024). There are other possible resolutions—e.g. some estimates include

substitutions of neutral alleles via hitchhiking. Nevertheless, it is curious that the empirical collapse of historical arguments for neutral theory has not yet led to a re-evaluation of related arguments by Haldane. Here, we revise Haldane's arguments for the modern era, finding them compatible with empirical evidence for abundant adaptation, while still imposing upper limits that might matter in some contexts.

Here, we first synthesize the historical literature, drawing out 2 key quantities (see Results). First, a population has a "reproductive excess", meaning how many individuals could be produced at a given life history stage, in excess of the minimum needed to maintain a constant population size, if all individuals had the best genotype present (see Glossary at end of article for definitions of the terms used here). The second key quantity is the proportion of deaths that are "selective deaths" (including foregone fertility), meaning deaths that causally contribute to changes in allele frequencies. The "cost of selection" is the number of selective deaths required for selection to effect the allele frequency changes needed to achieve a given adaptation rate.

Haldane's logic has previously been challenged (Felsenstein 1971; Maynard Smith 1968; Kern and Hahn 2018), most critically for his inappropriate reference to an optimal genotype that is unlikely to exist (Ewens 1970). However, Nei (1971) and Felsenstein (1971) derived a near-identical speed limit to Haldane's, without

this flaw, using a model that more explicitly captures the finite nature of reproductive excess. Here, we instead challenge Haldane's (1957) assumption that at most 10% of deaths could be selective, based on limited contemporary evidence on reproductive excess, given that this rough estimate was incorporated largely unchallenged into subsequent work.

Models of finite reproductive excess need at least 2 life history stages: adults and juveniles, where the latter shows reproductive excess relative to the former. Selective deaths are then relative not to a single adult population size $N$ as in Haldane's (1957) model, but to the population size at the appropriate life history stage (Kimura and Crow 1969). Once multiple life history stages are considered, assumptions regarding when and how density regulation operates must be made explicit (Nicholson 1933; Haldane 1956). Standard relative fitness models have an implied infinite number of juveniles subject to perfect density regulation in their transition to an adult population of size $N$ (Bertram and Masel 2019).

An emphasis on life history transitions rather than on generations is a strength rather than a weakness of the selective deaths view. One of the many flaws of the concept of "fitness" (Van Valen 1989) is the difficulty of defining a "generation" for many species, especially colonial species for which an "individual" is not well defined (Wilson and Barker 2021). Consider for example the budding yeast *Saccharomyces cerevisiae*. Is each mitotic division a generation? Or each life cycle spanning from meiosis to meiosis, with a variable number of mitoses in between? Or the span between outcrossing events, with variable occurrences of selfing as well as mitoses in between? Or is a generation best defined ecologically with respect to dispersal between resources that allow growth? Problems defining a generation arise for a broad range of species (albeit not humans, nor many other animal species), but are resolved when population dynamics are viewed as a series of life history transitions (Smith *et al.* 2024). The "generation" that matters in this view is not 1 complete life cycle, but rather the "generation" of reproductive excess, in contrast to other life history transitions that involve survival rather than reproduction.

After synthesizing the literature, here we reformulate and generalize Nei's (1971) and Felsenstein's (1971) ideas to selection on both fecundity and survival, to life cycles with selection at more than 1 stage, and to life cycles with a variable number of stages. We clarify the concepts of reproductive excess and selective deaths, and use our general theory to pose 2 empirically accessible questions. First, how much reproductive excess does the best genotype have? Second, what fraction of deaths can be selective, and how does this compare to the 10% limit assumed by Haldane? Posing questions in this form allows us to make the first empirical estimates with which to ground Haldane's approach. We use data from Exposito-Alonso *et al.* (2019), who counted or estimated every plant grown and seed produced of *Arabidopsis thaliana* cultivars from 517 different genotypes in 1 season, under 8 distinct environmental conditions. These data are not representative of natural conditions, but they suffice to illustrate how such an analysis can be done. Ours is the first direct application of selective death and reproductive excess arguments to empirical data.

## Results
### Synthetic historical review

Haldane's argument in his seminal 1957 paper contains 2 key parts, both novel at the time. In the first part, he defined "selective deaths" as the subset of deaths $s(1 - p)N$ that contribute to an increase in the allele frequency $p$, assuming a haploid population where $1 - s$ individuals without the beneficial allele survive to reproductive maturity for every 1 individual with the beneficial allele. These deaths are considered selective because the $(1 - p)N$ individuals that lack a beneficial mutation expect $s$ more deaths than they would if they had the mutation, and those extra deaths are required for selection to have its effects. Selective "deaths" can include absence of potential offspring, not just literal deaths, because reduced fecundity is mathematically equivalent to increased mortality.

Haldane defined the "cost of selection" as the number of selective deaths occurring during a substitution (i.e. a selective sweep from low allele frequency to fixation). He calculated this cost as the integral of $s(1 - p)N$ over the course of a sweep from allele frequency $p = p_0$ to close to 1 (Fig. 1a). In a haploid population of constant size $N$, 1 sweep requires $N \times D$ selective deaths, where $D = -\ln(p_0) + O(s)$. For appropriately small $s$ and $p_0$ (Haldane suggests $s < (1/3)$ and $p_0 = 10^{-4}$), the first term dominates, making $D$ nearly independent of $s$. For alternative assumptions about ploidy, dominance, and degree of inbreeding, $D$ is a different function of $p_0$, but $s$ remains unimportant unless close to 1 (Haldane 1957). $D = 18.4$ for a representative case of $p_0 = 10^{-4}$ at a diploid autosomal locus with fitnesses 1, $1 - s$, and $1 - 2s$ for individuals homozygotic for the beneficial allele, heterozygotic, and homozygotic for the wild-type allele, respectively. Haldane conservatively estimated that 20–30 $N$ selective deaths are needed per sweep in natural populations.

In the second part of his argument, Haldane converts this estimate of selective deaths per sweep into an upper bound on the number of sweeps per generation. Haldane considered independent sweeps at $x$ loci, such that the current allele frequency $p_i$ at the ith locus reduces population fitness by a factor of $1 - d_i = 1 - s_i(1 - p_i)$ relative to its post-sweep value. The fitness of the population is then lower than that of a hypothetical population fixed for all segregating beneficial alleles by a factor of $\prod_{i=1}^{x} (1 - d_i) \approx e^{-\sum_{i=1}^{x} d_i}$. Haldane observed that the mean individual therefore has $\sim e^{-\sum_{i=1}^{x} d_i}$ surviving descendants for every 1 surviving descendant that an individual with all segregating beneficial alleles would have. Assuming that this ideal individual exists, and that sites evolve independently, there are $\sum d_i N$ selective deaths per generation. Comparing these to the 30$N$ selective deaths needed per sweep, Haldane obtains a minimum average spacing between fixation events of $n \geq (30/\sum d_i)$ generations.

Haldane argues that a maximum "selection intensity" defines an upper bound on $\sum d_i$, and therefore on the speed of adaptation. Haldane defines selection intensity $I = \ln(s_0/S)$, where $s_0$ is the survival of the best genotype in the population and $S$ is the average survival of the population. Haldane's calculations proceed with the implicit assumption that the best genotype in the population (referenced by $s_0$) will also be the genotype with the beneficial version at all $n$ segregating sites. Plugging in $n = (30/\sum d_i)$ yields a selection intensity $I = \ln(s_O/S) = 30/n$.

Finally, Haldane argued that 10% selection intensity was approximately the upper limit that could be sustained long-term in most species, citing limited empirical knowledge of selection intensities available at the time. From this estimated value, he derived a speed limit based on $I \leq 0.1 = (30/n)$, yielding $n \geq 300$ generations between sweeps. Subsequent discussion remained framed around this 10% figure.

However, if the optimal genotype is not present in the population, selection intensity will be lower than Haldane calculated (Ewens 1970). An optimal genotype is likely present under stabilizing selection, which Haldane's (1954, 1957) use of infant survival

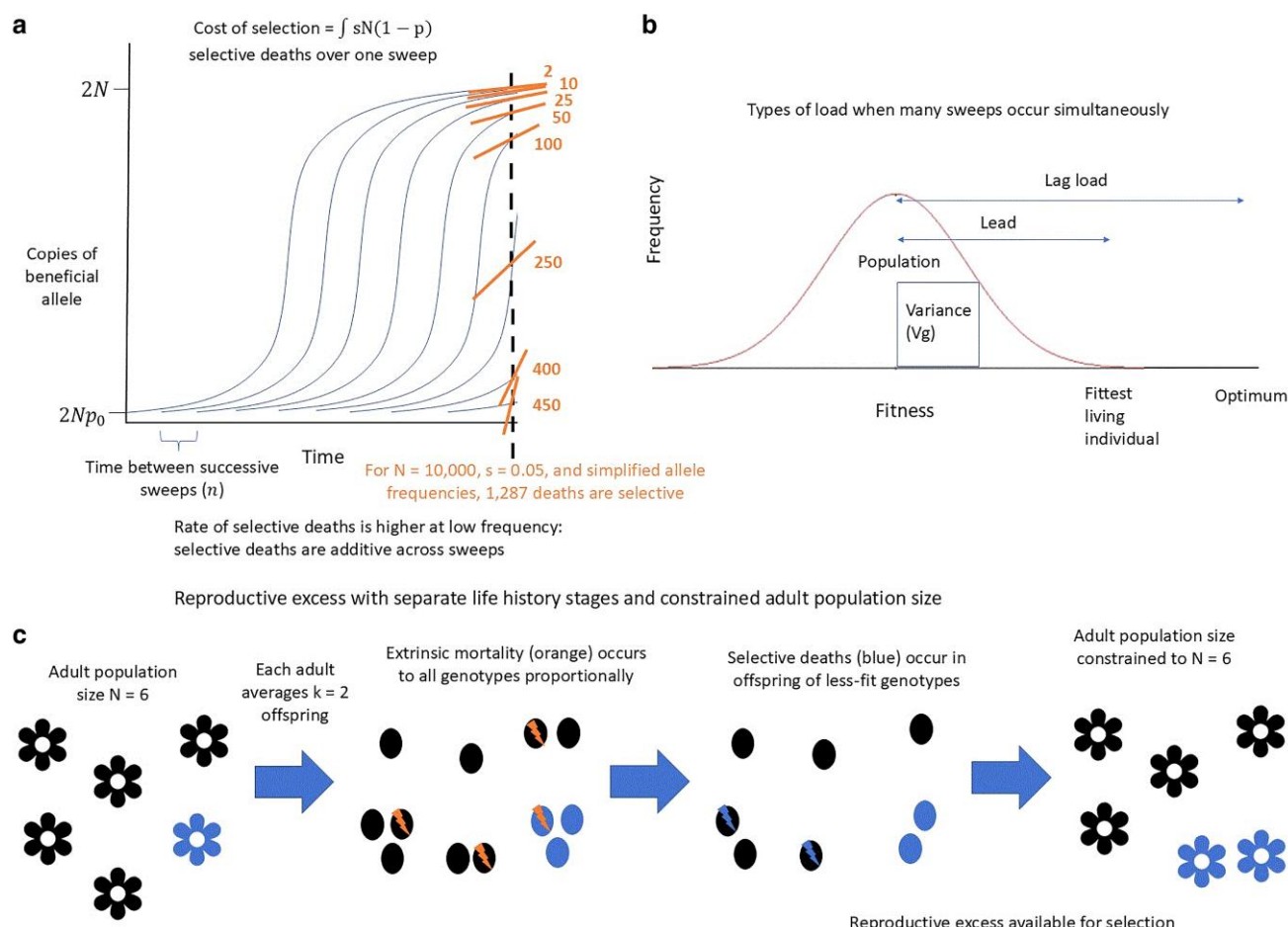

**Fig. 1.** Three different types of arguments have been used to argue for limits to the speed of adaptation. a) The cost of selection for 1 sweep is the cumulative number of selective deaths that must occur to complete that sweep (each independent sweep shown as a logistic curve). The cost of selection at 1 time point is the sum of the instantaneous costs $s(1 - p)N$ being incurred by each sweep at that time point, illustrated as the slopes of the short straight lines, calculated along the vertical dashed line. For this example, we calculate the total number of selective deaths for the set of allele frequencies $P = 0.05, 0.1, 0.5, 0.8, 0.9, 0.95, 0.98$, and $0.996$. b) Load is a normalized difference between an optimal fitness and mean population fitness: $L = (W_{opt} - \bar{W})/W_{opt}$. Different loads can be defined by comparing to different "optimal" fitnesses. Lag load uses the difference between a hypothetical optimal genotype and the population mean, while the lead uses the difference between the best currently living genotype and the population mean. c) Finite reproductive excess imposes an upper limit on how many selective deaths per generation are possible, which sets an upper bound to how fast substitutions can occur. Haldane (1957) combined the cost of selection with lag load to produce his speed limit, while Ewens (1970) corrected this calculation to use lead. In contrast, Nei (1971) and Felsenstein (1971) use the cost of selection and finite reproductive excess to derive a speed limit.

as a function of birth weight suggests he was envisaging. However, his calculations entail directional selection.

Today, "selection intensity" has acquired a somewhat different definition within the Breeder's Equation of quantitative genetics (Walsh and Lynch 2018). Haldane's (1957) selection intensity, defined with respect to fitness (infant survival) rather than phenotype (birth weight), is better compared to load $L = (W_{opt} - \bar{W})/W_{opt}$, where $\bar{W}$ is population mean fitness relative to some optimal genotype's fitness $W_{opt}$.

Load relative to the best genotype present is now known as the "lead", inspired by the degree to which the fittest part of the population "leads" the bulk (Desai and Fisher 2007) (see Fig. 1b, and the Glossary at end of article for definitions of the terms used here). We use the term "lag load" (Maynard Smith 1976) for load relative to an optimal genotype $W_{opt} = 1$ possessing all currently segregating beneficial alleles (see Fig. 1b and the Glossary at end of article for definitions of the terms used here).

The fact that there are so many amino acid substitutions, each requiring a sweep, was the original evidence supporting neutral theory

(Kimura 1968). Kimura and Ohta (1971) plugged in estimates of the actual rate of substitution in mammalian lineages as $n$, and rearranged Haldane's equation to get $L = e^{-(30/n)}$, which produced what they considered to be an excessively large load. Their argument was subtly different from Haldane's, arguing that a high load implies that typical individuals would need to have a biologically implausible fraction of their offspring die (Kimura and Ohta 1971). It propagates Haldane's (1957) problematic assumption that an individual with all beneficial variants is present in the population (Ewens 1970).

Ewens (1970) adjusted Haldane's (1957) calculations to instead use the lead. Prior to recent traveling wave models (Desai and Fisher 2007), approximations for the lead were derived from variance in fitness (Kimura 1969; Ewens 1970). With many independent sweeps at once, variance in fitness (after normalizing mean population fitness as 1) is approximately $s/n$, where $s$ is the selection coefficient of an adaptive allele and $n$ is the number of generations between fixation events (Ewens 1970). The fittest genotype likely to be present can be estimated using the statistics of extreme values, e.g. around 4.9 SD above the mean for a population

of size $10^6$ (Ewens 1970). Using Haldane's 10% as an upper bound on the lead instead of on the lag load yields $4.9\sqrt{s/n} = 0.1$. For $s = 0.01$, $n$ is around 20, much less than Haldane's estimate of 300, and $n$ is lower still for lower $s$. In other words, Ewens (1970) found that many simultaneous sweeps do not imply an implausibly large lead, and the corresponding speed limit of $n \approx 20$ is not an obstruction with respect to observed rates of amino acid divergence. Similar arguments have been applied to deleterious mutation load (Galeota-Sprung *et al.* 2020).

Although Ewens' argument revolved around lead, which is a normalized difference between fitnesses, his approach continued the traditional emphasis of evolutionary genetics on variance in fitness, which describes the mean square of differences (Fisher 1930; Crow 1958; Ewens 2004). Modern traveling wave theory instead derives the lead directly from $s$, $N$, and the beneficial mutation rate $U$, and obtains the variance in fitness variance only downstream from that (Desai and Fisher 2007), rather than relying on our ability to directly measure fitness and its variance as an input to the calculation of the lead.

Maynard Smith (1968) made a quite different argument against speed limits, claiming that the reason Haldane's dilemma is not a problem is pervasive synergistic epistasis. Synergistic epistasis increases differences in fitness above those expected from differences in the numbers of beneficial mutations, thereby making each selective death tend to count toward a larger number of sweeps at once. A persistent source of confusion has been that Maynard Smith modeled epistasis using truncation selection, where the least fit half of a population dies each generation, with the explicit assumption that the remaining half always have enough reproductive capacity to reconstitute the population (Maynard Smith 1968). The degree to which the relaxed limits on adaptation in Maynard Smith's model can be attributed to epistasis vs assuming infinite reproductive excess (see below) is thus unclear.

Although Ewens' lead-based approach negates the second part of Haldane's arguments that converted lag load into a speed limit via selection intensity, this does not negate the first part of Haldane's argument, namely that 20–30 $N$ selective deaths are needed for a sweep. Nei (1971) and Felsenstein (1971) made a key advance building on just this first part of Haldane's argument. This advance has not received substantial attention, perhaps partially due to confusion about terminology, and also because growing acceptance of the neutral theory of molecular evolution had seemed to resolve the immediate issues posed by speed limits.

The cost of selective deaths must be paid out of a finite budget of reproductive excess. The relative fitness models that dominate population genetics (e.g. Wright–Fisher and Moran) implicitly assume inexhaustible reproductive excess (Bertram and Masel 2019). This can be seen easily when simulating selection on zygote viability using rejection sampling—when fitness is low, an absurd number of zygotes might be generated and discarded prior to filling the $N$ slots, potentially enabling the few survivors to include exceptionally rare beneficial events such as double mutations. However, real populations have a finite reproductive excess, e.g. human females do not easily give birth to more than 20 infants. This constrains members of the next generation to options within a finite set of potential offspring. This concept has been applied to lethal mutagenesis strategies for antiviral drugs (Bull *et al.* 2007).

Nei (1971) and Felsenstein (1971) each modeled independently evolving sites in a haploid population with adult population size $N$. Each adult has fecundity $k$, i.e. produces $k$ offspring prior to juvenile deaths (Fig. 1c, first arrow). In their deterministic models, $k$ is exact, but the theory readily generalizes to interpreting $k$ as an expectation. The raw reproductive excess is thus $(k-1)N$, with

$k > 1$. Extrinsic mortality that occurs prior to selective mortality, and hence at a fixed rate, can be treated as an effectively lower value of $k$ (Fig. 1c, second arrow). Haldane's 10% selection intensity estimate is equivalent to 10% of deaths being selective, which corresponds to $k = 1.1$. Nei (1971) and Felsenstein (1971) retain this 10% cap for illustrative purposes, where $k$ includes the fact that extrinsic mortality substantially reduces the fecundity available to be "used" for selective deaths. Population size regulation after selective mortality (Fig. 1c, far right) will on average consume the remaining reproductive excess that was not used by selection.

The population undergoes sweeps, all with the same initial frequency $p_0$ and selection coefficient $s$ applying to survival rather than fecundity. Each sweep follows the same trajectory with a mean delay of $n$ generations between sweeps (Fig. 1a). Given independent sites, the cost of selection is summed across loci at any given point in time (e.g. slopes of short, straight lines in Fig. 1a); Haldane calculates the expectation of this sum by integrating across time points for a single sweep. Comparing this cost to the reproductive excess of the population produced the novel result that the minimum spacing $n$ is $-\ln(p_0)/\ln(k)$ (Felsenstein 1971; Nei 1971). For Haldane's estimates of $p_0 = 10^{-4}$ and $k = 1.1$, this yields $n = 97$ generations between selective sweeps. This can be compared to Haldane's original spacing of $-\ln(p_0)/\ln(W_{max}/\bar{W}) = 92$, where Haldane's selection intensity $\ln(W_{max}/\bar{W}) = 0.1$ (by assumption) has been replaced by $\ln(k) = \ln(1.1) \approx 0.095$.

Importantly, this speed limit calculation is not subject to the same criticisms as Haldane's original argument. Where Haldane's calculation includes a comparison between the mean fitness of the population and an optimal genotype which may or may not exist, Nei's (1971) and Felsenstein's (1971) approach compares the available reproductive excess to the reproductive excess required to effect changes in allele frequencies. Even if no individual exists who possesses the beneficial allele at every segregating site, each sweep still requires a certain fraction of deaths to contribute to its selection. It is the finite nature of reproductive excess that directly produces this limit on the rate of adaptation.

While an identical speed limit is derived by both papers, they interpret the speed limit differently. Nei (1971) interprets $k$, and hence the speed limit, in the context of a strongly density-regulated population of constant size. Under this interpretation, the rate at which mutations sweep, each affecting relative fitness, is constrained, but this cannot threaten population persistence. Felsenstein (1971) considers the upper bound on $k$ applying at low population density and applies this to the context of ongoing environmental change that degrades absolute fitness, requiring adaptive rescue at a rate constrained by a speed limit. A population that lacks sufficient reproductive excess, even at low density, goes extinct.

Critical questions remain unanswered. Felsenstein (1971) considers a hypothetical population at the edge of viability, while Nei (1971) considers a hypothetical population adapting at the maximum possible speed—is it possible to relate their key theoretical quantities to fecundity and mortality data on real populations? How do the speed limit and its consequences depend on the details of density regulation in real populations, and the timing of selection relative to density regulation? Do populations with high fertility have substantially more permissive speed limits, or is this balanced by greater loss of reproductive excess to extrinsic mortality?

## Theory
### *Life histories compatible with Nei's and Felsenstein's model*
Life history goes through at least 2 transitions, which in Nei (1971) and Felsenstein (1971) are simple: births and deaths, in that order.

They assign all genotypes the same fertility $k$. They consider only the limiting case in which all deaths are selective. Some violations of these obviously unrealistic assumptions can be incorporated without changing the model.

Nonselective deaths can be incorporated as causing $k$ to fall to $k_{effective}$, where Nei (1971) and Felsenstein (1971) solve for Cost of selection = $k_{effective} - 1$. Density-independent nonselective deaths, aligned with Felsenstein's (1971) interpretation, are well modeled as occurring prior to selection on survival, reducing $k$ in the current generation. Density-dependent nonselective deaths, aligned with Nei's (1971) interpretation, are best incorporated into the framework after selection on survival. Here, they can balance out deaths that were "unused" by selection in the current generation, by reducing $k$ in the subsequent generation. Figure 1c illustrates both kinds of nonselective deaths, proceeding first through nonselective fecundity to produce raw reproductive excess, then the non-density-dependent component of extrinsic mortality, then selective deaths, and finally density-dependent extrinsic mortality to cap the population size at $N$ adults.

Nei (1971) and Felsenstein (1971) emphasize literal deaths, but were presumably aware that differences in fecundity also contribute selective "deaths". This is because mathematically, foregone fecundity is equivalent to deaths that take place immediately after fecundity, and can be treated as:

$$\text{Selective ``deaths" during differential fecundity} = N_i(b_{best} - b_i)$$

where $N_i$ is the number of reproductive mature adults, $b_i$ is the fecundity of genotype $i$, and $b_{best}$ is the fecundity of the genotype with the highest fecundity in that environment.

### Reproductive excess within a fixed life cycle

Next, we generalize from just 1 life history transition experiencing selection, to multiple that occur in a consistent order. We consider a life history transition $j$ that starts with population size $N_j$ and ends at population size $N_{j+1} = k_j N_j$: We now define:

$$\text{Reproductive excess after transition } j = k_j N_j - N_{min,j+1}$$

where $N_{min,j+1}$ is the minimum population size at the end of transition $j$ that is required in order for the population to achieve size $N_j$ at the beginning of transition $j$ in the next life history cycle. Note that $k_j > 1$ indicates fecundity, while $k_j \leq 1$ indicates survival.

For example, consider a microbial population in batch culture that alternates between exponential growth, stationary phase, and dilution life history transitions. For stationary phase, $k$ is not much less than 1 but $kN$ is relatively large because $N$ is large. Many cells will be lost in the subsequent dilution (say 1:100), but then exponential growth will typically yield $N$ cells from any starting density, so long as seeding is successful. $N_{min} > 100$ is thus required to recover $N$ live cells at the beginning of the next stationary phase. In contrast, for the dilution life history transition, $k \sim 0.01$ and $N_{min}$ required in principle to regrow the population in permissive media may not be much larger than 1.

Reproductive excess for transition $j$ provides the maximum budget for how many selective deaths could occur during transition $j$, without upsetting population dynamics by overwhelming the compensatory capacity of density regulation during other transitions. The calculation of $N_{min,j+1}$ includes density regulation at all life history transitions except $j$, while density regulation during transition $j$ is held constant in its representation by $k_j N_j$. Reproductive excesses specific to a particular life history transition already take other transitions into account, and so reproductive excesses across different transitions do not need to be further combined.

To produce selective deaths, $k_{i,j}$ must depend on genotype $i$. To capture density regulation, $k_{i,j}$ for at least some values of $j$ must depend on population size $N_j$. The values $k_{i,j}$ can also be functions of the genotype frequencies and/or an absolute measure of time. Two life history transitions (survival and fecundity) are the minimum, but each of these can be broken up into multiple transitions. For example, survival ($k < 1$) can be broken into components representing survival at different ages, or into a selective component depending only on genotype vs a density-dependent extrinsic mortality component depending only on $N_j$ vs an extrinsic mortality component occurring at a constant rate. The degree to which selection and density regulation act on births vs deaths has consequences, e.g. for rapid adaptation (Draghi *et al.* 2024).

Reproductive excess is calculated for a focal population. The best choice of focal population depends on the question that the model is intended to answer. For example, studying balancing selection calls for the actual population, while studying evolutionary rescue (Bell 2017) in an asexual population calls for a hypothetical population where each individual has the current best genotype present. These choices will result in different values of $k_j$ and $N_{min,j+1}$.

To calculate reproductive excess with respect to the actual population, we solve for $N_{min,j+1}$ in:

$$N_j = N_{min,j+1} \sum_i f_i \prod_{x \neq j} k_{i,x},$$

where $f_i$ is the frequency of genotype $i$ at the beginning of the transition. With respect to a population where each individual has the best genotype, we instead solve for $N_{min,j+1}$ in:

$$N_j = N_{min,j+1} \prod_{x \neq j} k_{best, x}.$$

### Speed limits with multiple life history stages

Under Nei's (1971) interpretation, a population has only 1 value of reproductive excess RE = $(k - 1)N$, and 1 (adult) population size $N$, from which he derived a speed limit of $-\ln((RE/N) + 1/\ln(p_0))$ substitutions per generation. When selective deaths occur at only 1 life history transition $j$, this expression applies with $RE_j$ and $N_j$.

However, because each selective death reduces $k$ for the transition at which it occurs, it reduces reproductive excess at all other transitions, in a manner that depends on assumptions about when and how density regulation occurs. E.g., the selective death of an adult tree creates a canopy opening into which saplings may grow, in addition to reducing the number of seeds available to grow into it. Both these density regulation effects dampen the degree to which selective deaths at 1 life history transition reduce reproductive excess at other transitions. Reproductive excess calculations invoke $N_{min}$ as a counterfactual—how many losses at the focal life history transition could density regulation compensate for, holding all else constant? There is no simple counterfactual for perturbing selective deaths at multiple transitions at once, even when each invokes distinct, non-pleiotropic mutations. Life history trade-offs make matters worse, because the same allele might cause selective deaths at multiple transitions, even when the allele frequency does not change from 1 life cycle to the next. We thus reach the negative result that there is no general formula for a speed limit when selection occurs during more

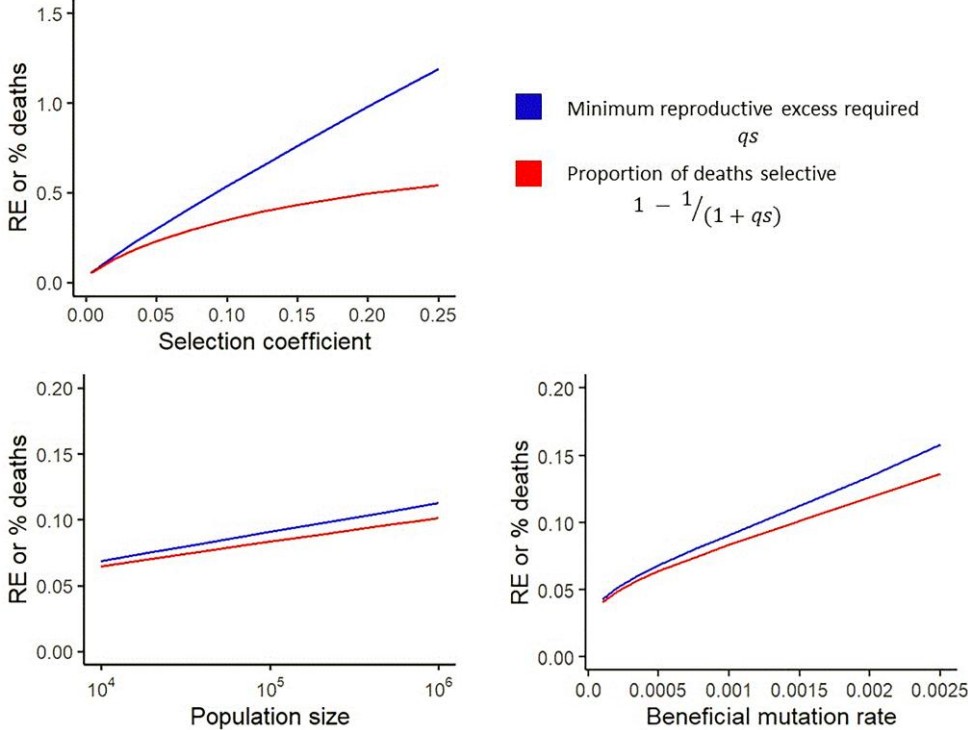

**Fig. 2.** As the adaptation rate goes up with increasing $U$, $s$, and $N$, following the multiple mutations regime of Desai and Fisher (2007), so do the proportion of deaths selective and the minimum reproductive excess required to sustain that rate of adaptation. Where not otherwise specified, $U = 0.001$, $s = 0.01$, and $N = 10^6$. Parameter ranges are truncated to avoid the regime $(s/U) < 3$, for which the assumptions of the Desai and Fisher's (2007) model break down. The minimum required reproductive excess $qs$ and the corresponding proportion of selective deaths $1 - 1/(1 + qs)$ were calculated by numerically solving Equation 39 for $q$ in Desai and Fisher (2007). We note that Desai and Fisher (2007) use relative Malthusian fitness $r = r' - \bar{r}$, rather than per-generation $w = W/\bar{W}$, but results converge for small $s$.

than 1 life history transition—it depends on the details of density regulation.

### Reproductive excess beyond a fixed life cycle

Not all organisms proceed through the same sequence of life history transitions every cycle, e.g. budding yeast experience a variable number of mitoses in between each meiosis, and a variable number of selfing events between each outcrossing. In this case, we cannot take the product of an exact series of transitions. Instead, we privilege the life history transition that produces the most severe bottleneck, assuming that the population will spring back to vibrancy after. We define a minimum number of individuals $N_{bot}$ who need to make it through to the other side of the bottleneck, and define:

Reproductive excess at transition $j = k_j N_j$
  – min. needed to ensure $N_{bot}$ after bottleneck.

We now need to take the expectation over all possible series of life history transitions, and solve for $N_{min,j+1}$ in:

$$N_{bot} = E\left( N_{min,j+1} \prod_{x}^{\substack{\text{life history stages} \\ \text{between } j \text{ and bot}}} k_{best, x} \right).$$

The precise value of $N_{bot}$ will be informed by the ecology of the species in question. It may be small, such as when just a modest number of new hosts, each colonized by just 1 infectious

microorganism, are sufficient to ensure the population's future. The appropriate value of $N_{bot}$ is the smallest population size that reliably escapes extinction.

### Comparison to fitness

Haldane obtained selective deaths from $sN(1 - p)$, where $s$ is the selection coefficient with respect to per-generation relative fitness. Our $k$ values are equivalent to absolute fitness components per life history transition. Selective deaths can be derived from an underlying population dynamic model, without requiring either generation or relative fitness to be defined first. E.g., in the simple case of an absolute fitness version of the Wright–Fisher model, adults of genotype $i$ produce an expected $k_i$ offspring, then die. Selective deaths per capita $k_{best} - k_{mean}$ are equal to the difference in absolute fitness. Total population reproductive excess is $N k_{mean} - N_{min}$. By definition, $N = N_{min} \times k_{best}$. Substituting $(N/k_{best})$ for $N_{min}$ and then dividing by $N$ gives the per-capita reproductive excess $= k_{mean} - (1/k_{best})$.

However, these simple relationships with fitness break down once selection occurs at multiple life history stages. When a genotype that benefits fitness in 1 life history transition bears an antagonistically pleiotropic cost at another, costs and benefits cancel out. In contrast, selective deaths never cancel out; indeed, incomplete density compensation causes selective deaths to accrue across life history transitions. Similarly, there is no canceling out across generations, e.g. seasonally fluctuating selection must incur many selective deaths to effect the large allele frequency fluctuations that have been observed around the long-term mean (Machado et al. 2021; Kelly 2022; Rudman et al. 2022).

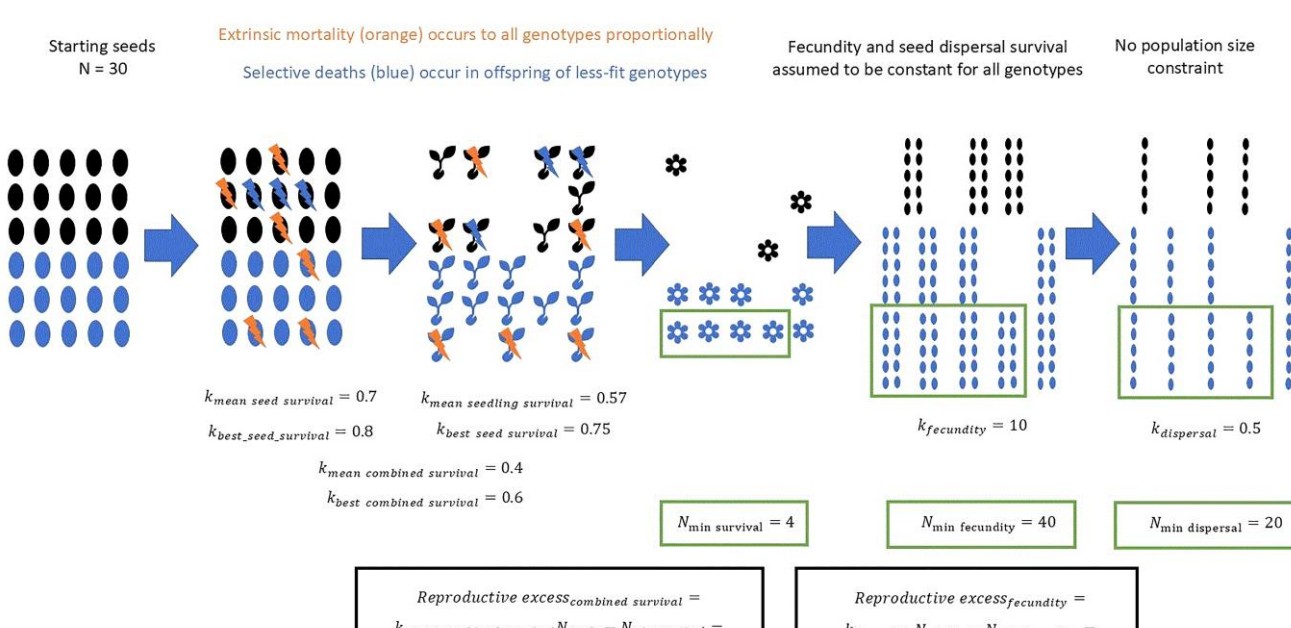

**Fig. 3.** Worked example of reproductive excess for the life history transitions of A. *thaliana* in the experimental setup from Exposito-Alonso *et al.* (2019). We found no evidence for between-genotype differences in fecundity and the experimental setup provides no information about seed dispersal, so we show no selective deaths during these transitions. Values of *k* are chosen for illustrative purposes.

Discrete time models such as Wright–Fisher typically emphasize the effective reproductive number rather than the Malthusian parameter, i.e. equivalent to $b/d$ rather than $b - d$ in a continuous time model with births at rate $b$ and deaths at rate $d$. Malthusian approaches are generally used to handle the complications arising from multiple life history transitions (Charlesworth 1994). Our approach enables an effective reproduction number framing in these more complex scenarios, while still avoiding the need to define 1 "generation". This enables treatment of the finite nature of reproductive excess.

### Comparison to traveling wave models

Desai and Fisher's (2007) Eq. 39 solves for the fitness difference or "lead" $qs$ between the highest-fitness "nose" with $r_{max} = 1 + qs$ and the average genotype $\bar{r} = 1$ in an asexual population, as a function of $U$, $N$, and relative fitness difference per mutation $s$. The minimum reproductive excess of the best genotype that is implicitly required to maintain constant population size is equal to the lead $qs$. This aligns closely with the parameter $k - 1$ of Nei (1971) and Felsenstein (1971). The actual reproductive excess in Desai and Fisher's (2007) model is infinite, as for all models that assume a constant population size and treat only relative fitness.

We next calculate the proportion of deaths that are selective. The entire reproductive output of the best genotype present, $1 + RE \geq 1 + qs$, represents nonselective deaths (or foregone fecundity) among its offspring. Other genotypes all experience the same rate of nonselective deaths. The per-capita odds that the next death hits the offspring of a specific average individual rather than a specific nose individual are $1:1/(1 + qs)$. Consider a time interval in which an individual with the best genotype expects $1+$ RE nonselective offspring deaths. An average parent expects $(1 + RE)(1 + qs)$ offspring deaths during this time, $1 + RE$ of which are nonselective. This makes the proportion of deaths that are selective equal to $1 - 1/(1 + qs)$.

For a substantial range of parameters, especially with rapid adaptation with large $s$, $U$, and $N$, both the minimum required reproductive excess, and the proportion of juvenile deaths that are selective, exceed the value of 0.1 that was previously assumed for both (Fig. 2). This application to the model of Desai and Fisher (2007) helps clarify the relationship between these 2 properties. In this particular model, with its explicit adults and implicit juveniles under selection, they are both functions of the lead $qs$, but this need not be so simple when more complex life histories are considered. We note that the adaptation speed has a one-to-one relationship with the proportion of deaths that are selective, whereas reproductive excess imposes a speed limit without having a direct effect on speed.

## Experimental proof of concept

We measure reproductive excess and the proportion of deaths that are selective for the first time in an empirical system using data from Exposito-Alonso *et al.* (2019) (Fig. 3). Briefly, 517 homozygous lines of A. *thaliana* were grown in $2 \times 2 \times 2$ environments of varying location, water availability, and adult density. See Supplementary Methods, Supplementary Tables 1 and 3, and Supplementary Figs. 1–5 for more details. For high density, 30 seeds of the same genotype were planted per pot, and selective deaths are for the transition from seed to adult. For low density, ~10 seeds were planted per pot, and only 1 seedling, chosen at random, was retained after germination; selective deaths are for the transition from seedling to adult. Fitness differences in fecundity were not significant (see Supplementary Results).

The proportion of deaths that are selective exceeds 10% (Fig. 4, *x*-axis, Supplementary Table 2). In Madrid with low water and high density, as many as 95% of deaths were selective.

A priori, we expect high water and low density to be more benign, which might increase reproductive excess and/or reduce

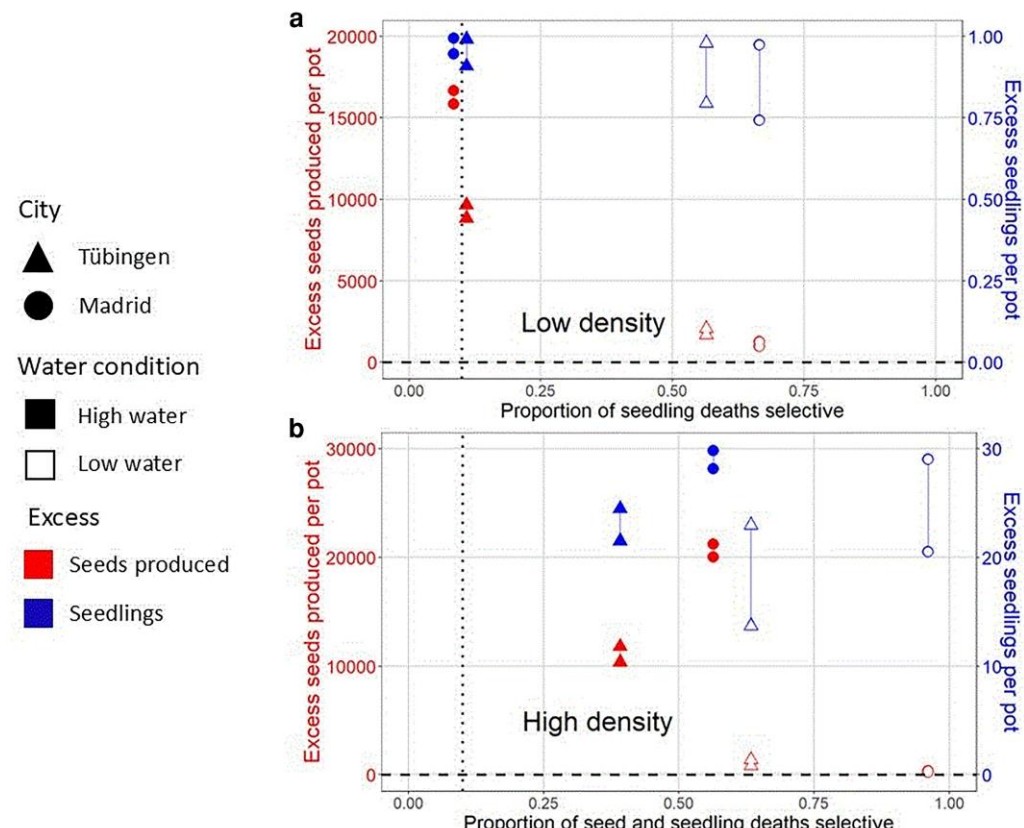

**Fig. 4.** The proportion of deaths that are selective generally exceeds Haldane's 10% estimate (dotted vertical line), with ample reproductive excess (above 0, shown as dashed horizontal line) especially under high-water conditions. Selective deaths shown on the *x*-axis apply either to seeds plus seedlings a) or to seedlings alone b). Reproductive excess is calculated (see Supplementary Information) for 2 life history transitions: seeds produced and seedlings surviving to adulthood. These are shown in red and blue, and correspond to values shown on the left and right *y*-axes, respectively. Note that reproductive excess of seedlings cannot exceed 1 per pot under low density conditions, and 30 per pot under high density conditions. Reproductive excesses are shown as a vertical range representing high and low values of seed survival during dispersal, a life history transition for which we lacked empirical data, and therefore considered a 10-fold plausible range. The proportions of deaths that are selective are adjusted for extreme value bias as shown in Supplementary Table 1. Values can be found in Supplementary Table 2.

## Discussion

the proportion of deaths selective. While we cannot compare seedling deaths at low density to seed plus seedling deaths at high density, these predictions are confirmed for high vs low water: strongly in the case of selective deaths and excess seeds, and weakly with respect to excess seedlings (Fig. 4, Supplementary Table 2). Estimated reproductive excess is fairly insensitive to the 10-fold range we consider for the proportion of seeds that successfully disperse to suitable habitats (vertical line length in Fig. 4).

## Discussion

Population geneticists no longer exhibit obvious concern that Haldane's Dilemma poses a limitation to the speed of adaptation, even when e.g. reporting high rates of adaptive sweeps (Sella *et al.* 2009; Galtier 2016; Uricchio *et al.* 2019; Murga-Moreno *et al.* 2024). Here, we began by reviewing the flaw in load-based speed limit arguments; with appropriate comparisons to the best genotype currently present, rapid adaptation is possible. However, there is a different and crucial type of limitation pointed out by Nei (1971) and Felsenstein (1971), one that depends not on relative load but on the finite nature of reproductive excess. Given an assumed 10% limit on the fraction of deaths that could be selective, and allowing selection at just 1 life history transition, the rate of adaptation is significantly limited (Felsenstein 1971; Nei 1971). We clarified and extended theoretical arguments regarding selective

deaths drawn from finite reproductive excess, to apply even to life histories for which a "generation" is poorly defined, e.g. colonial organisms. Speed limits depend on when and how density regulation occurs, and our proof-of-principle empirical analysis suggests that the fraction of selective deaths may not be limited to 10%; these results might explain why adaptation is not significantly limited by reproductive excess.

Nei (1971) and Felsenstein (1971) solved for a hypothetical equilibrium where all "effective" reproductive excess was converted to selective deaths. They thus demonstrated that finite reproductive excess imposed a speed limit, without referring to lag load. Our more complete treatment emphasizes that reproductive excess varies by life history stage, and that the degree to which selective deaths during 1 life history transition reduce reproductive excess at other transitions depends on how and when density regulation occurs. While these theoretical results do not provide a general expression for a speed limit to adaptation, they uphold the previous special case, which is compatible with some populations in some cases having limits on their rate of adaptation.

We apply the concepts of reproductive excess and selective deaths to empirical data for the first time. This proof-of-principle application was sufficient to suggest a novel possibility: Haldane's 10% limit to selective deaths may be too low. The smallest proportion of selective deaths we observed across 8 environmental conditions was 8.5%, while in the more adverse environmental

conditions within this experiment, 95% of deaths were selective. Relaxing this auxiliary assumption about a critical parameter value resolves Haldane's concerns. Future work under more natural conditions (e.g. with higher extrinsic mortality) and in different species (e.g. less fecund) remains necessary to reach the conclusion that the proportion of deaths that are selective is typically high. Our theoretical framework is flexible enough to be customized for any species, using whichever life history transitions best describe that species' life history. Uses might include understanding the limits to adaptation given high fecundity (Hutchings 2000). We note that the method of Ewens (1970) might allow estimates of $V_G$ for per-generation fitness, combined with population size, to be repurposed to estimate the proportion of deaths selective.

Empirical demonstration of these concepts makes the concepts more concrete. One contribution of our current work is simply to clarify the variety of lines of reasoning that produce limits on the rate of adaptation. Our more specific theoretical and empirical analyses then develop a line of reasoning about reproductive excess and selective deaths that has been neglected. The attention of "creation science" to the question of Haldane's dilemma (Remine 2005, 2006) highlights the importance of clearly resolving it.

Different papers use the term "substitutional load", which we avoid here, to mean very different things, and the resulting confusion in terminology has obscured the consequences of formulating Haldane's dilemma in different ways. "Substitutional load" has been used to refer to what we here call the lag load (Kimura and Ohta 1971; Maynard Smith 1976), the cost of selection (Kimura 1968), the number of offspring that the most fit genotype must produce (Ewens 2004), the sum of lag load across all generations involved in a substitution (Kimura 1960; Nei 1971), and even more broadly to refer to variance-based rather than load-based arguments when made in the context of similar questions (Ewens 1970). The latter highlights standard modes of reasoning in genetics, especially quantitative genetics, which follow Fisher (1930) to focus on variances—sums of differences squared—while selective deaths and reproductive excess are, like load, both differences, with no square operation.

The optimal genotype re lag load is normally defined, as Haldane did, with respect to the best genotype that can be constructed from segregating alleles. This is because a standard relative fitness model makes environmental change implicit. When fitness is defined in absolute terms, the optimal genotype used to define lag load can also include adaptive mutations that have not yet appeared in the population (Bertram et al. 2017; Osmond and Klausmeier 2017). While speed limits cannot threaten the persistence of a species adapting in a constant environment, real species face rapidly changing environments (biotic and abiotic) that can threaten population persistence. Even in a model of absolute fitness, large lag load need not be a problem, although it must be stable rather than growing. I.e., for a population to persist, adaptation must keep up with the speed of environmental change.

It is interesting that Ewens' (1970) key concept of relative load was later reinvented as the "lead", as part of calculations that derived the actual speed of adaptation $v$ (rather than limits to it) from the beneficial mutation rate $U$, the population size $N$, and the per-mutation selection coefficient $s$ (Desai and Fisher 2007). One reason this solution was not available to Haldane or Ewens was that population genetics had not yet begun to treat origination processes (McCandlish and Stoltzfus 2014). Instead of treating a steady input of beneficial new mutations, Haldane considered a scenario in which environmental change activates beneficial variants within standing genetic variation. Indeed, a variant's initial frequency $p_0$ is the primary factor in determining

Haldane's maximum speed of adaptation. Some adaptation comes not from activation of standing genetic variation, but from de novo mutations each appearing at initial frequency $1/N$ or $1/2N$. A lead-based approach was used for de novo mutations to derive the rate of beneficial sweeps in asexuals as $(2s \ln[Ns]/\ln^2[s/U])$ for the simple case of constant $s$ (Desai and Fisher 2007), for parameter ranges in which the previously derived rate $UNs$ does not apply. Here, we relaxed the assumption of infinite reproductive excess made by this relative fitness model, and calculated the minimal reproductive excess required for the model to hold. We also reveal the model's implied fraction of deaths (or foregone fecundity) that are selective. Both quantities are functions of the lead.

Yet, another approach, starting with Kimura (1961), is to use the framework of information theory to set bounds on the speed of adaptation. Natural selection increases the information stored in genomes (Adami 2012). Kimura calculates the amount of information acquired per sweep in terms of $p_0$ and then relates this to the cost of selection using Haldane's equation that $D = -\ln(p_0)$ (Kimura 1961). More recent approaches treat the bounds placed on the accumulation of information in much more detail, while treating either the Malthusian parameter (McGee et al. 2022) or classic discrete time relative fitness (Hledík et al. 2022). Both these approaches define an information "cost", but this is not equal to our cost in terms of selective deaths.

Like Haldane (1957), Nei (1971), and Felsenstein (1971), we have neglected complexities from linkage disequilibrium, which will typically make the conversion of selective deaths to adaptation less efficient. Here, we considered both the independent sites extreme, and the other extreme in the form of the asexual model of Desai and Fisher (2007). Other limits to adaptation speed can be imposed by genetic map length (Weissman and Barton 2012; Neher 2013; Weissman and Hallatschek 2014).

Excitingly, unlike most approaches in evolutionary genetics, the approach we describe does not require the quantity "fitness", which is deceptively difficult to define in a manner that can be generalized to all circumstances (Van Valen 1989; Ariew and Lewontin 2004; Doebeli et al. 2017; Bertram and Masel 2019; Smith et al. 2024). Standard quantitative definitions of either relative or absolute "fitness" require a clear definition of a "generation" over which change is assessed, which in turn requires a clear definition of an "individual" whose life cycle a generation captures (Wilson and Barker 2021). Our generalized selective deaths approach is derived from selection but does not require "fitness" to be defined. Rather, we measure selective deaths from pairwise differences in fecundity and survival between each genotype and the best genotype present, during each life history transition. Reproductive excess corresponding to that life history transition indicates how stringent selection can be, without triggering a decline in population size. This approach applies at each life history transition and can therefore be generalized to species with complex life histories, where it becomes difficult to define a "generation" and therefore fitness.

## Glossary

Absolute fitness of an adult: expected number of individual offspring surviving to adulthood (defined here for hermaphrodite species—half this value, if reproducing sexually with males).

Absolute fitness of a juvenile: expected number of individual offspring (or half this value, if reproducing sexually with males). Failure to survive to adulthood (reproductive maturity) implies 0 offspring.

Adult: reproductively mature individual. More than 1 adult life history stage may be defined.

Cost of selection: the number of selective deaths that must occur over time to accomplish defined evolutionary change, e.g. to complete a single selective sweep.

Individual: an organism that meets a loosely defined set of criteria (Wilson and Barker 2021), including a shared genome, and the degree of integration of parts. Whether e.g. a group of microbes is a closely connected ecological community vs an individual organism may be a matter of biological judgment.

Juvenile: individuals that are not yet reproductively mature. More than 1 juvenile life history stage may be defined, e.g. before vs after dispersal.

Lag load: $(W_{optimal} - \bar{W})/W_{optimal}$ where $W_{optimal}$ is a theoretical optimal genotype that might not be present in the population and $\bar{W}$ is the average genotype present.

Lead: $(W_{optimal} - \bar{W})/\bar{W}$ where $W_{optimal}$ is the best genotype present in the population and $\bar{W}$ is the average genotype present.

Life history transition: survival (i.e. persistence of an individual), reproduction (i.e. generation of new individuals), and/or organismal growth from 1 life history stage to the next.

Load: a difference in fitness between an actual genotype or population and a reference, generally divided by some reference fitness. See "lag load" and "lead" as concrete examples.

Relative fitness: expected relative genetic contribution to the next generation.

Reproductive excess: the degree to which a hypothetical population of the optimal genotype would conclude a life history transition with a larger population than the minimum required to complete a life history cycle without the population shrinking in size.

Selective deaths: the subset of deaths (or foregone fertility) that contributes to selective changes in allele frequency. This can be quantified as how many deaths each genotype experiences that would not have been experienced if that genotype were replaced by the best genotype.

## Data availability

Data for the proof of concept analysis is available at Figshare (Exposito-Alonso and Weigel 2018a, 2018b). All analysis was performed in R, and our code is available on GitHub: www.github.com/josephmatheson/selective_deaths.

Supplemental material available at GENETICS online.

## Acknowledgments

We thank Taylor Kessinger for sparking the last author's original interest in this topic many years ago by presenting Haldane's (1957) paper in journal club, and Jason Bertram, Ryan Gutenkunst, and Ulises Hernández for helpful discussions. We thank Joe Felsenstein for his helpful review of a previous version of this manuscript, which helped us understand the differences in interpretation between his paper and Masatoshi Nei's, as well as Sally Otto and 3 other anonymous reviewers.

## Funding

This research was supported by the John Templeton Foundation [62028] and [62220]. Open Access publication was financially supported by the John Templeton Foundation through the Consortium for Advancing a Science of Purpose at the University of Minnesota. The opinions expressed in this article are those of the authors, and not those of the John Templeton Foundation.

## Conflicts of interest

The author(s) declare no conflicts of interest.

## Author contributions

Initial conception of the paper by JMas and ME-A, with multiple rounds of reconception by JMat and JMas. Design and interpretation by JMat and JMas. Data and details of experimental methodology provided by ME-A. Data analysis mostly by JMat, with the MCMCglmm model analyzed by ME-A. Manuscript first drafted by JMat and substantively revised many times by all authors.

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

*Editor: S. Otto*