## [Peer Review File · Genetics]

Substitution load revisited: a high proportion of deaths can be selective

Joseph Matheson and Joanna Masei

NOTE: The reviews and decision letters are unedited and appear as submitted by the reviewers.

In extremely rare instances and as determined by a Senior Editor or the EIC, portions of a review may be redacted. If a review is signed, the reviewer has agreed to no longer remain anonymous.

The review history appears in chronological order.

Review Timeline:

Submission Date:	2024-07-31
Editorial Decision:	2024-09-03
Resubmission Received:	2024-10-04
Editorial Decision:	2024-10-23
Resubmission Received:	2024-12-16
Accepted:	2024-12-22

EDITORIAL NOTE: Reviews of this manuscript were transferred to GENETICS from another journal. Decision letters and reviews from outside of GENETICS have been redacted from the Peer Review History document to remove material for which no permission to publish was obtained.

August 20, 2024

GENETICS-2024-307328

Substitution load revisited: a high proportion of deaths can be selective

Dear Dr. Matheson:

Two experts in the field have reviewed your manuscript, and I have read it as well. While your manuscript is not currently acceptable for publication in GENETICS, we would welcome a substantially revised manuscript. Both reviewers have comments and concerns to be addressed in a revised manuscript. You can read their reviews at the end of this email.

Based on the previous round of reviews and my own assessment (below), it is my opinion that the article still needs major revision in order to be a strong contribution for GENETICS. That said, I think that the authors are absolutely correct that now is a fantastic time to revisit the cost of substitution and its meaning in light of recent theoretical and empirical work. Rather than taking more time for review, I suggest the authors take this opportunity to address the points made below.

We also think that this piece may be better as a "Perspectives article." While there is new material, it is largely exploratory, pushing the theory to consider a broader swath of life histories. While I do not insist on this point, I do think that a Perspectives would better match the paper.

We look forward to receiving your revised manuscript. Please let the editorial office know approximately how long you expect to need for revisions.

Upon resubmission, please include:

1. A clean version of your manuscript;
2. A marked version of your manuscript in which you highlight significant revisions carried out in response to the major points raised by the editor/reviewers (track changes is acceptable if preferred);
3. A detailed response to the editor's/reviewers' feedback and to the concerns listed above. Please reference line numbers in this response to aid the editor and reviewers.

Your paper will likely be sent back out for review.

Additionally, please ensure that your resubmission is formatted for GENETICS
<https://academic.oup.com/genetics/pages/general-instructions>

Follow this link to submit the revised manuscript: Link Not Available

Sincerely,

Sarah Otto
Associate Editor
GENETICS

Aneil Agrawal
Senior Editor
GENETICS

Approved by:
Howard Lipshitz
Editor in Chief
GENETICS

Associate Editor Comments:

The paper is a welcome treatment of the cost of natural selection and the limits those costs impose on the rate of adaptation, in light of more modern theoretical and empirical work. That said, there were a few issues that I remained unconvinced about.

* Much is made in this manuscript about the "ideal genotype" in Haldane's second argument (e.g., line 117). Nowhere in

Haldane (1957) does the word "ideal" appear, and I read Haldane's intent slightly differently. On pp. 520-521, Haldane relates the cost of natural selection to the intensity of selection, "I". Both in the breeding literature and in Haldane (1954), "I" is a measure referring solely to genotypes within the current population, comparing attributes of surviving individuals with the whole population. It is the current value of "I" that Haldane seeks, regardless of what is ideal. Specifically, on p. 512, Haldane defines $I = \ln(s_0/S)$ where "for any range of phenotypes there is a phenotype with optimal survival, s_0 , compared with S in the whole population" and then goes on to give examples where it is clear that s_0 refers to types within the population (not some hypothetical optimum or ideal that does not exist). To estimate the intensity of selection, Haldane then gives a hypothetical scenario where fitness is half what it could be in a new environment ($I = \ln 2 = 0.69$) and argues that this is likely on the high end, relative to empirical estimates the intensity of selection. So while the halving of fitness in this scenario might be seen as relative to an ideal that does not exist, what Haldane aims for is the current intensity of selection among existing genotypes, which he relates to the rate at which sweeps are occurring and causing selective deaths.

Note that the Haldane's estimate of $I=30/n$ holds when only one selected locus is segregating at a time (where the "ideal" genotype should exist), and he showed, approximately, that it would hold with several loci too on p. 521. Again, I don't think it is the existence of the best genotype that matters so much as the imposition of a certain intensity of selection in the present.

I can definitely see where the authors can read Haldane as focusing on a distant non-existent optimal genotype, but I don't think that was Haldane's intent. This also opens up another way to empirically assess Haldane's argument: what are the ranges of empirically measured intensities of selection? Haldane took 0.69 to be on the large end (approximately that seen in *Biston betularia*), but he also pointed out that $I=0.03$ for human birth weights, so he took $I=0.1$ to be a more "probable value", leading to $n \sim 300$. How does this probable value compare to intensities of selection measured to date? [Perhaps available in the database of Kingsolver et al. 2001?]

* To my reading, the tone is more dismissive of previous thinkers, including Haldane, than necessary. I suggest revising because it was pretty prescient for Haldane to consider what the limits to selection might be. By a slight change in wording, the paper can be reframed as the early questions raised about the limits of selection remain an important (if somewhat confused) topic, and you seek to place the cost of natural selection into the context of more recent theoretical and empirical research. (Examples of unnecessarily negative tone, in my opinion, are in the specific comments below.)

* Although the manuscript nicely contrasts the approach of Haldane to that of Desai and Fisher for an asexual population (I really liked this!), limits to adaptation have also been studied for sexual populations. Relating the limits of selection in sexuals would be of great interest (see Neher AREES 2013 for an older review; Weissman and Barton PLoS Genetics 2012; Weissman and Hallatschek 2014 Genetics), although those limits are imposed by the genetic map length, not by reproductive limits. Considering both might be a great future project.

SPECIFIC COMMENTS

Line 50: In "excess of requirements" needs description at first use (to replace the existing population size, so above one offspring per parent)

Line 55: "without evidence" makes it sound like Haldane made a bad choice ignoring evidence at the time, whereas there really was little evidence that he could use. "lacking much evidence at the time" would be more accurate.

Line 58: "Haldane (1957) confused" is not generous enough, in my opinion, to what Haldane accomplished. Yes, he alluded to optimal genotypes, but his main goal, in my opinion, was to relate the current intensity of selection (measurable among the genotypes currently present) to the number of selective sweeps per unit time (n).

Line 95: "muddying the waters from the beginning". I agree with the previous reviewer that his raising two key points was part of the same argument, and it was not muddying the waters to make them at the same time.

Lines 120-121: I don't think this reflects Haldane's logic on pp. 520-521 (in particular, he doesn't assume "N deaths available"), unless I misunderstand your intent

Line 125: "even better optimal genotype" -> "even better genotype through epistatic interactions" [Epistasis does not change which of all possible genotypes is optimal. Please reword.]

Line 130: "which he incorrectly equated with 10% lag load" As noted above, he equated 10% with the intensity of selection, not lag load, and he did have some limited empirical evidence that he cited.

Line 145: "named it" -> "named this difference"

Line 145 & 151: The definition of "lead" is ambiguous. Here and in the figure, it is defined as a difference in the fitnesses between the average phenotype and the best phenotype currently present, but on line 151 it is used as a load (i.e., measured relative to some fitness measure, maybe the fittest genotype present?).

Line 183: "might therefore be due to..." Why either or? I would imagine both epistasis and the definition of load would both influence the result.

Line 204: Vague, maybe "can be treated as an effectively lower value of k"

Line 218: I don't think it was odd for Haldane to describe this as a selection intensity. It was the intensity (I) of current selection that he was after, relating this to the rate of sweeps and the number of deaths per sweep.

Line 247: Is there a "-1" missing. $k=1$ should correspond to no room for a cost of selection for population persistence (from Felsenstein's perspective).

Line 256: My sense is that Nei, Felsenstein, and Hamilton spoke of literal deaths to help make their point clearer, not because they didn't recognize differences in fecundity could also contribute. I agree that this point deserves attention, but I wouldn't phrase this as they didn't realize this point (see, e.g., page 176 in Nei).

Line 272: 1/100 dilution -> dilution (say 1/100)

Line 275: "may not be much larger than 1" is not true in practice in batch culture (it is true when plating down to single colonies). Rephrase so it doesn't seem so at odds with typical batch culture designs?

Lines 290-294: This phrasing of reproductive excess didn't make sense to me. This is not how "reproductive excess can be calculated" but rather how it can be related to selective deaths?

Line 316: Could be clearer. That is, wouldn't the allele frequency change between the life stages? Drop the last phrase about allele frequencies or clarify.

Line 302 & 320: The reader could use more sign posting about what is considered in these sections. Section on line 302 is said to discuss a general model, but then the section on line 320 indicates that this wasn't a general model.

Line 430: I'm not sure that this is the prevailing consensus. Reference?

Line 442: "adaptation is not significantly limited" - I don't think this claim is generally true, and it certainly requires references to support it. The fact that evolutionary rescue is not assured even when it is possible is one example where adaptation is significantly limited. The limits to selection explored (e.g., in Weissman and Hallatschek 2014 Genetics) allow unbounded reproductive excess, but still find strong limits to adaptation placed on sexual organisms by recombination.

Line 454: "while in the most adverse environmental conditions" - One could argue that low water was not as challenging as many natural environments, so that more environmentally induced deaths would occur in wild populations (limiting the amount of "reproductive excess available for selection").

Line 476-478: I agree that this isn't really novel. Drop?

Lines 502-520: Consider moving to the introduction. This framing really helps motivate the paper and cast a fresh eye on the substitution load.

Line 507: substitutions -> amino acid substitutions (right?)

Figure 1: Panel a - what is "simplified allele frequency", is this just $1-p$? What is "modified for extrinsic mortality" ("Average number of survivors $k_{\text{post}} = 1.3$ " would be clearer here)

Figure 2: What were the other parameter values used in these panels?

Figure 4: It is hard to tell which points correspond to the y-axis on the left and on the right (hard to see what is hollow and what is solid). Maybe colour code by y axis?

Associate Editor Comments:

The paper is a welcome treatment of the cost of natural selection and the limits those costs impose on the rate of adaptation, in light of more modern theoretical and empirical work. That said, there were a few issues that I remained unconvinced about.

* Much is made in this manuscript about the "ideal genotype" in Haldane's second argument (e.g., line 117). Nowhere in Haldane (1957) does the word "ideal" appear, and I read Haldane's intent slightly differently.

Haldane discusses the “optimal genotype”, rather than “ideal”. We did not intend those two terms to read differently and have switched to “optimal”.

On pp. 520-521, Haldane relates the cost of natural selection to the intensity of selection, "I". Both in the breeding literature and in Haldane (1954), "I" is a measure referring solely to genotypes within the current population, comparing attributes of surviving individuals with the whole population.

We disagree that “intensity” in the breeding literature refers to genotypes. Intensity in the breeding literature compares the mean phenotype of selected individuals to the mean phenotype of the population. By referring to phenotypes rather than breeding values, it is not equal to the intensity one would obtain in reference to genotypes. In contrast, Haldane’s intensity does apply to genotypes in practice in his calculations, even if he uses the word “phenotype”; the corresponding “phenotype” is “fitness”, and there is no phenotype defined as only partially heritable.

In modern terms, Haldane’s intensity, which is calculated with respect to fitness rather than with respect to phenotype, is equal to the lead of Desai & Fisher 2007, rather than to the selection intensity of the modern breeding literature.

It is the current value of "I" that Haldane seeks, regardless of what is ideal. Specifically, on p. 512, Haldane defines $I = \ln(s_0/S)$ where "for any range of phenotypes there is a phenotype with optimal survival, s_0 , compared with S in the whole population" and then goes on to give examples where it is clear that s_0 refers to types within the population (not some hypothetical optimum or ideal that does not exist).

Haldane certainly did not intend to refer to a nonexistent optimal genotype, he intended to refer to the current best genotype. We have revised language throughout the introduction and historical review to clarify this intent on Haldane’s part.

To estimate the intensity of selection, Haldane then gives a hypothetical scenario where fitness is half what it could be in a new environment ($I = \ln 2 = 0.69$) and argues that this is likely on the high end, relative to empirical estimates the intensity of selection. So while the halving of fitness in this scenario might be seen as relative to an ideal that does not exist, what Haldane aims for is the current intensity of selection among existing genotypes, which he relates to the rate at which sweeps are occurring and causing selective deaths.

The criticism made by Ewens 1970 is that when a genotype's fitness is half what it could be, but the genotype with twice its survival doesn't exist, the intensity of selection among existing genotypes does not equal 0.69. We try to make clearer where in Haldane's logic this gap between his intent and mathematical practice opened up (see next response below).

Note that the Haldane's estimate of $I=30/n$ holds when only one selected locus is segregating at a time (where the "ideal" genotype should exist), and he showed, approximately, that it would hold with several loci too on p. 521. Again, I don't think it is the existence of the best genotype that matters so much as the imposition of a certain intensity of selection in the present.

Haldane's estimate holds with one locus. His iconic calculation (the second and third paragraph of the Discussion in Haldane 1957), considers i loci, each depressing fitness by d_i , with fixations in the whole population occurring at a rate of one every n generations. He says that fitness is reduced from 1 to $e^{-\sum d_i}$, $n = 30/\sum d_i$, the difference between the fittest genotype and the average $= \frac{s_0}{s} = e^{-30/n}$, and $I = \ln\left(\frac{s_0}{s}\right) = 30/n$. As Ewens pointed out first, this assumes that s_0 is the genotype with all segregating beneficial alleles and is also the phenotype with optimal survival currently in the population. This genotype almost certainly does not exist in the population, and so Haldane's calculation of $I = 30/n$ will not be correct.

We have clarified this logic in the text and more clearly attributed the criticism to Ewens (1970).

I can definitely see where the authors can read Haldane as focusing on a distant non-existent optimal genotype, but I don't think that was Haldane's intent.

We have modified language to make clearer that we do not think Haldane was intentionally focusing on a nonexistent genotype.

This also opens up another way to empirically assess Haldane's argument: what are the ranges of empirically measured intensities of selection? Haldane took 0.69 to be on the large end (approximately that seen in *Biston betularia*), but he also pointed out that $I=0.03$ for human birth weights, so he took $I=0.1$ to be a more "probable value", leading

to $n \sim 300$. How does this probable value compare to intensities of selection measured to date? [Perhaps available in the database of Kingsolver et al. 2001?]

As discussed above, selection intensities are with respect to phenotypes, whereas our calculations require the much more difficult estimation of fitness, which is enabled by the extraordinarily rich Arabidopsis dataset, with many replicates of the same genotype in the same environment.

Instead of reporting the lead (which is Haldane's "selection intensity"), we report the closely related proportion of deaths which are selective (see relationship in Figure 2). The latter more directly relates to the limit to adaptation rate derived by Felsenstein and Nei (see calculation in orange in Figure 1A). While the two are very similar, the key difference is that the lead refers to Malthusian fitness, where the proportion of deaths which are selective refers to lifetime fitness. We have added a sentence to make this relationship clear.

To calculate the proportion of deaths which are selective, we need to be able to estimate the fitness of the best genotype present, including correction for extreme value bias, and we need mean population fitness. Fitness must be in per-generation rather than in Malthusian terms. There may be more suitable datasets out there, but we didn't find easily them from Kingsolver et al. 2001, which focused on phenotypic data rather than fitness. We added a sentence to the Discussion highlighting the potential to use Ewens' approach to repurpose suitable V_G data on per-generation fitness to obtain the proportion of deaths selective, albeit not reproductive excess.

* To my reading, the tone is more dismissive of previous thinkers, including Haldane, than necessary. I suggest revising because it was pretty prescient for Haldane to consider what the limits to selection might be. By a slight change in wording, the paper can be reframed as the early questions raised about the limits of selection remain an important (if somewhat confused) topic, and you seek to place the cost of natural selection into the context of more recent theoretical and empirical research. (Examples of unnecessarily negative tone, in my opinion, are in the specific comments below.)

We completely agree that Haldane's 1957 paper was prescient on this topic, and we did not at all intend to come across dismissive of Haldane! Previous rounds of feedback and revision on this manuscript were focused on resolving points of confusion about the specific nature of Haldane's arguments and which parts were subject to later criticism from Ewens and others. The negative tone may have crept in during that process.

We have revised to more clearly emphasize the historic contributions of Haldane and others, and express the importance of questions surrounding the cost of selection. This includes better attribution of criticism of Haldane's arguments to previous thinkers, especially Ewens, rather than make the criticism seem to be coming from us.

* Although the manuscript nicely contrasts the approach of Haldane to that of Desai and Fisher for an asexual population (I really liked this!), limits to adaptation have also been

studied for sexual populations. Relating the limits of selection in sexuals would be of great interest (see Neher AREES 2013 for an older review; Weissman and Barton PLoS Genetics 2012; Weissman and Hallatschek 2014 Genetics), although those limits are imposed by the genetic map length, not by reproductive limits. Considering both might be a great future project.

This is a very interesting topic, we agree. Meantime, we briefly refer to this relevant work in the Discussion.

SPECIFIC COMMENTS

Line 50: In "excess of requirements" needs description at first use (to replace the existing population size, so above one offspring per parent)

We have reworded as "*in excess of the minimum needed to maintain a constant population size*". Note that reproductive excess, in our final scheme, can also be defined for life history stages other than reproduction.

Line 55: "without evidence" makes it sound like Haldane made a bad choice ignoring evidence at the time, whereas there really was little evidence that he could use. "lacking much evidence at the time" would be more accurate.

Language changed – Haldane's guess was indeed informed by the limited evidence available at the time. We do not believe he intended for his guess to remain unchallenged for so long!

Line 58: "Haldane (1957) confused" is not generous enough, in my opinion, to what Haldane accomplished. Yes, he alluded to optimal genotypes, but his main goal, in my opinion, was to relate the current intensity of selection (measurable among the genotypes currently present) to the number of selective sweeps per unit time (n).

Changed both to clarify Haldane's intent and emphasize the groundbreaking nature of Haldane's initial calculation. We do agree with Ewens that despite Haldane's intent, an optimal genotype is implicit in one part of his calculation, and we now make clearer where that was.

Line 95: "muddying the waters from the beginning". I agree with the previous reviewer that his raising two key points was part of the same argument, and it was not muddying the waters to make them at the same time.

We removed this phrase.

Lines 120-121: I don't think this reflects Haldane's logic on pp. 520-521 (in particular, he doesn't assume "N deaths available"), unless I misunderstand your intent

We have substantially reworked this portion of the paper to expand on Haldane's logic, beginning with the paragraph that previously ended with these lines. The phrase "N deaths available" is not necessary and has been removed during the re-write.

Line 125: "even better optimal genotype" -> "even better genotype through epistatic interactions" [Epistasis does not change which of all possible genotypes is optimal. Please reword.]

This was not our intended meaning, and we have substantially rewritten this paragraph on lag load to be clearer.

Line 130: "which he incorrectly equated with 10% lag load" As noted above, he equated 10% with the intensity of selection, not lag load, and he did have some limited empirical evidence that he cited.

This paragraph has been deleted, and replaced with earlier more comprehensive coverage of Haldane's logic.

Line 145: "named it" -> "named this difference"

This sentence was deleted as a result of introducing the lead earlier.

Line 145 & 151: The definition of "lead" is ambiguous. Here and in the figure, it is defined as a difference in the fitnesses between the average phenotype and the best phenotype currently present, but on line 151 it is used as a load (i.e., measured relative to some fitness measure, maybe the fittest genotype present?).

Lead is the difference in fitness between the average genotype and the best genotype present, not the average phenotype and the best phenotype present. This makes lead a kind of load.

We do not refer to phenotypes in our manuscript, only to genotypes and fitness.

Line 183: "might therefore be due to..." Why either or? I would imagine both epistasis and the definition of load would both influence the result.

We have edited to acknowledge that both might matter, with unknown relative contributions.

Line 204: Vague, maybe "can be treated as an effectively lower value of k"

Changed.

Line 218: I don't think it was odd for Haldane to describe this as a selection intensity. It

was the intensity (I) of current selection that he was after, relating this to the rate of sweeps and the number of deaths per sweep.

We have removed this phrase.

We also now introduce Haldane's selection intensity metric much earlier in the paper and at much greater length.

Line 247: Is there a "-1" missing. $k=1$ should correspond to no room for a cost of selection for population persistence (from Felsenstein's perspective).

Added the -1, thank you for catching that!

Line 256: My sense is that Nei, Felsenstein, and Hamilton spoke of literal deaths to help make their point clearer, not because they didn't recognize differences in fecundity could also contribute. I agree that this point deserves attention, but I wouldn't phrase this as they didn't realize this point (see, e.g., page 176 in Nei).

Changed phrasing to emphasize that Nei and Felsenstein were presumably aware of this.

Line 272: 1/100 dilution -> dilution (say 1/100)

Changed.

Line 275: "may not be much larger than 1" is not true in practice in batch culture (it is true when plating down to single colonies). Rephrase so it doesn't seem so at odds with typical batch culture designs?

N_{\min} here refers to the minimum number theoretically required to recover the population. Batch culture in liquid media is normally designed to transfer far more than a single cell, but it would in theory only require one or a very small number of cells to make it into the next flask in order to grow up to a full population. We have rephrased to make clearer that N_{\min} is distinct from the actual number of cells transferred during dilution.

Lines 290-294: This phrasing of reproductive excess didn't make sense to me. This is not how "reproductive excess can be calculated" but rather how it can be related to selective deaths?

We intended this sentence to express the fact that reproductive excess must be calculated for a focal genotype/population, which the modeler must choose for their purpose. We have reworded.

Line 316: Could be clearer. That is, wouldn't the allele frequency change between the life stages? Drop the last phrase about allele frequencies or clarify.

Reworded to clarify that allele frequency doesn't change from one whole life cycle to the next.

Line 302 & 320: The reader could use more sign posting about what is considered in these sections. Section on line 302 is said to discuss a general model, but then the section on line 320 indicates that this wasn't a general model.

Subsection heading changed to more specifically indicate that the section on speed limits is treating speed limits with more than one life history stage.

Line 430: I'm not sure that this is the prevailing consensus. Reference?

We have no reference. We have occasionally heard Haldane's dilemma brought up, often by older members of our community, e.g. following talks concerning a high proportion alpha of amino acid substitutions being adaptive. The answers often given when this question arises perhaps more precisely indicate that the community is no longer concerned, to the point of forgetting about Haldane's dilemma, rather than that there is a clear consensus. We have reworded, somewhat hobbled by the lack of clear literature-based ways to back up our impressions from oral interactions. We instead cite with respect to the absence of discussions of Haldane's literature in an area where one would expect to find them.

Line 442: "adaptation is not significantly limited" - I don't think this claim is generally true, and it certainly requires references to support it. The fact that evolutionary rescue is not assured even when it is possible is one example where adaptation is significantly limited. The limits to selection explored (e.g., in Weissman and Hallatschek 2014 Genetics) allow unbounded reproductive excess, but still find strong limits to adaptation placed on sexual organisms by recombination.

Clarified that this statement applies only to limits from reproductive excess. Other limits may apply.

Line 454: "while in the most adverse environmental conditions" - One could argue that low water was not as challenging as many natural environments, so that more environmentally induced deaths would occur in wild populations (limiting the amount of "reproductive excess available for selection").

We have weakened "most" to "more" and clarified that this comparison is within the experiment.

Line 476-478: I agree that this isn't really novel. Drop?

We have removed this phrase.

Lines 502-520: Consider moving to the introduction. This framing really helps motivate the paper and cast a fresh eye on the substitution load.

We have moved this material to the Introduction.

Line 507: substitutions -> amino acid substitutions (right?)

Changed.

Figure 1: Panel a - what is "simplified allele frequency", is this just $1-p$? What is "modified for extrinsic mortality" ("Average number of survivors $k_{\text{post}} = 1.3$ " would be clearer here)

Clarified in the figure legend that this phrase means that allele frequencies were chosen arbitrarily for the purpose of simplified calculations.

Figure 2: What were the other parameter values used in these panels?

We added these to the figure legend.

Figure 4: It is hard to tell which points correspond to the y-axis on the left and on the right (hard to see what is hollow and what is solid). Maybe colour code by y axis?

Changed as suggested. We have also increased the size of the text and data points in the figure and made hollow points more visibly hollow.

October 23, 2024

GENETICS-2024-307426

Substitution load revisited: a high proportion of deaths can be selective

Dear Dr. Masel:

Thank you for your revision. While the revisions have substantially improved the paper, the core of the introduction remains too unclear to me as a representative reader to recommend acceptance at this stage (see comments below).

In particular, clarification is needed at these points:

* It is my opinion that pages 7-9 will confuse the reader rather than clarify the limits of adaptation. This might be because the text has been updated in parts but not everywhere, so it seems to go back and forth between discussing the best genotype present and the best genotype possible with the alleles present.

* The definition of load is non-standard. When speaking of fitnesses (W), the load is measured relative to a standard (dividing by other W_{\max} or W_{mean} , Crow 1970) and is scale-free, but the paper defines loads only as differences (see glossaries). That might be because the reference is taken to have fitness one, but this is not clear. It may be that the authors intend to define load in terms of Malthusian fitness (m), but that is not what the reader would assume. The load in Desai and Fisher is defined by those authors in terms of Malthusian fitnesses, but this is also not clear to the reader of this manuscript. Specifying some formulae along the way, might help.

In my opinion, a substantially clarified introduction is needed for this paper to be suitable in GENETICS.

If you feel that my assessment is incorrect, I could send it out for review, but I was hoping to reduce the load on reviewers by addressing the points raised previously.

Upon resubmission, please include:

1. A clean version of your manuscript;
2. A marked version of your manuscript in which you highlight significant revisions carried out in response to the major points raised by the editor/reviewers (track changes is acceptable if preferred);
3. A detailed response to the editor's/reviewers' feedback and to the concerns listed above. Please reference line numbers in this response to aid the editor and reviewers.

Additionally, please ensure that your resubmission is formatted for GENETICS

<https://academic.oup.com/genetics/pages/general-instructions>

Follow this link to submit the revised manuscript: Link Not Available

Sincerely,

Sarah Otto
Associate Editor
GENETICS

Approved by:
Howard Lipshitz
Editor in Chief
GENETICS

Associate Editor Comments:

Line 118: Specify the definition of s used here, e.g., "using a haploid model with fitness of $1+s$ for a genotype bearing the beneficial mutation relative to the wildtype, with fitness of one." (Unclear at first use whether fitnesses are $1-s:1$ or that this assumes a haploid population or equivalent selection in diploids or what.)

Line 123: with no dominance -> with additive alleles and heterozygous fitness of $1+s/2$ [for some, especially medical geneticists,

no dominance means $h=0$] Also, there are two possible interpretations of the fitness regime here $\{1, 1+s/2, 1+s\}$ or $\{1, 1+s, 1+2s\}$, so specify which one is being used.

Page 7&8 - This needs to be reworked. I found it very hard to follow, as it switches between alternative reference frames, and I suspect many readers will struggle. Some of the issues are highlighted below.

Line 133: There remains confusion in the definitions of the intensity of selection. The phrase "where s_0 is the survival of the best genotype in the population" matches Haldane's use (e.g., in his example with birth weights, where s_0 is considering the survival of the best phenotype in the current population), but then the next line switches to W_{max} , as the "genotype that has the beneficial version at all n segregating sites."

Again, I disagree with that interpretation of Haldane (see also Haldane 1954), but I think that this can be dealt with plainly. You could point out that Haldane focused on quantitative traits (like birth weight) when defining intensity of selection (both here and in 1954), arguing that there is often a fitness peak for the quantitative character, which "is not usually, if ever an extreme value, and is often close to the mean" (p. 481, Haldane 1954). However, when calculating the substitution load, Haldane must switch from considering a quantitative trait subject to stabilizing selection to fitness, which is subject to directional selection. As Ewens (1970) showed, Haldane's calculations are not consistent with using a reference genotype that is the best in the current population but rather the best genotype possible given the alleles currently circulating, which typically does not exist.

Line 145 - This isn't clear without reading Desai and Fisher. When stating that the lead is "equal to the fitness difference between the mean individual and the best genotype present", it would be natural to use the definition on line 134 of fitness (W as "absolute survival"), but Desai and Fisher were working with Malthusian fitnesses (where $m_{MEAN} - m_{BEST} = \log(W_{MEAN}/W_{BEST})$). Also, shouldn't it be the "fitness difference between the best genotype present and the mean" (lead is positive, right?).

Line 146 - Now the selection intensity seems to have flipped back to referring to the "best genotype present" (not the best genotype possible).

Line 149 - Define lag load explicitly here (it's unclear given the back-and-forth about how to interpret Haldane above)

Line 151 - "is normally defined as Haldane did, with respect to the best genotype that can be constructed from segregating alleles" Again, I don't think that was what Haldane did, but it might be what Haldane's calculations implicitly did (Ewens 1970).

Line 208 - Haldane's absolute fitness? The selection intensity is relative ($\ln(s_0/s)$), so even if absolute survival rates were discussed, a shift from absolute to relative fitness is immaterial? Clarify line 208.

Line 246 - "Where Haldane compared the mean fitness of the population to the mean fitness of a hypothetical population" - Did he? Where?

Line 364 - I couldn't follow this. Where does $k_{mean} - (1/k_{best})$ come from? Is this fraction formatted correctly?

Lines 561-566 - As noted by Crow (1970, Genetic Loads and the Cost of Natural Selection), load is defined *relative* to a reference fitness in the denominator (otherwise multiplying all fitnesses by some arbitrary constant c changes the load, which it shouldn't). Make the definitions precise (e.g., with formulae along the way) to clarify what you intend to be measuring as Malthusian fitnesses (as in Desai and Fisher for lead) or W 's.

Thank you for your revision. While the revisions have substantially improved the paper, the core of the introduction remains too unclear to me as a representative reader to recommend acceptance at this stage (see comments below).

In particular, clarification is needed at these points:

* It is my opinion that pages 7-9 will confuse the reader rather than clarify the limits of adaptation. This might be because the text has been updated in parts but not everywhere, so it seems to go back and forth between discussing the best genotype present and the best genotype possible with the alleles present.

We have substantially edited this section to improve clarity (we hope). We especially focused in our revisions on explicitly specifying what genotype is being referred to at different places in the text (including when both are referred to), and who or what is doing the referring (us as writers, Haldane as a writer, or a conceptual entity like 'lead' or 'lag load').

* The definition of load is non-standard. When speaking of fitnesses (W), the load is measured relative to a standard (dividing by other W_{\max} or W_{mean} , Crow 1970) and is scale-free, but the paper defines loads only as differences (see glossaries). That might be because the reference is taken to have fitness one, but this is not clear. It may be that the authors intend to define load in terms of Malthusian fitness (m), but that is not what the reader would assume. The lead in Desai and Fisher is defined by those authors in terms of Malthusian fitnesses, but this is also not clear to the reader of this manuscript. Specifying some formulae along the way, might help.

We are now more precise about which type of load (or fitness) is being discussed and explicitly give the mathematical definitions of each (including appropriate denominators) in the text and in the legend of Figure 1B. We thank the editor for adding clarity and precision to this section. Following Haldane, we usually divide by W_{\max} rather than W_{mean} , although we realize that Desai & Fisher normalize relative to r_{mean} , and we now use explicit equations to make that clearer in the appropriate spot.

In the historical literature we discuss, load is defined in the standard way relative to a per-generation W_{opt} , including in Haldane 1957 (although he doesn't use the term load in that paper). We now explicitly clarify this and the connection between Haldane's selection intensity and lead. As a mathematical convenience, Desai and Fisher characterize the lead in terms of Malthusian fitness m or r , which we now clarify, but our primary definition of lead in connection with Haldane is in terms of W . We now initially define the lead conceptually in a way that transcends the exact mathematical form, allowing it to be subtly different when discussing Haldane 1957 than when discussing Desai & Fisher.

In general, we treat load primarily to clarify historical confusion over Haldane's argument, while in the rest of the paper we focus on selective deaths and reproductive excess exclusively, without reference to load. The fact that we emphasize the nature of load as a difference is primarily to contrast with approaches based on variance.

Associate Editor Comments:

Line 118: Specify the definition of s used here, e.g., "using a haploid model with fitness of $1+s$ for a genotype bearing the beneficial mutation relative to the wildtype, with fitness of one." (Unclear at first use whether fitnesses are $1-s:1$ or that this assumes a haploid population or equivalent selection in diploids or what.)

We now explicitly define the form of selection: haploid, where $1-s$ individuals without the beneficial allele survive for every 1 individual with the beneficial allele.

Line 123: with no dominance \rightarrow with additive alleles and heterozygous fitness of $1+s/2$ [for some, especially medical geneticists, no dominance means $h=0$] Also, there are two possible interpretations of the fitness regime here $\{1, 1+s/2, 1+s\}$ or $\{1, 1+s, 1+2s\}$, so specify which one is being used.

Replaced the phrase 'no dominance' with an explicit definition of fitnesses as $\{1-2s, 1-s, 1\}$, following Haldane 1957.

Page 7&8 - This needs to be reworked. I found it very hard to follow, as it switches between alternative reference frames, and I suspect many readers will struggle. Some of the issues are highlighted below.

This section is dense with terminology and deceptively similar concepts. We hoped that Figure 1 would help clarify the alternative frames of reference and terminology, and besides the changes to specific sections discussed below, we have also added more in-text references to Figure 1 where appropriate. We have also rearranged the order of material in a way that we hope helps.

Line 133: There remains confusion in the definitions of the intensity of selection. The phrase "where s_0 is the survival of the best genotype in the population" matches Haldane's use (e.g., in his example with birth weights, where s_0 is considering the survival of the best phenotype in the current population), but then the next line switches to W_{max} , as the "genotype that has the beneficial version at all n segregating sites."

We have removed the reference to W_{max} .

We now clarify that Haldane's calculations implicitly assume that "the best genotype in the population" and "the genotype that has the beneficial version at all n segregating sites" are the same. Both here and in Haldane, "the best genotype in the population" is the appropriate referent of selection intensity. The confusion lies only in whether or not "the genotype that has the beneficial version at all n segregating sites" will be identical to "the best genotype in the population".

Again, I disagree with that interpretation of Haldane (see also Haldane 1954), but I think that this can be dealt with plainly. You could point out that Haldane focused on quantitative traits (like birth weight) when defining intensity of selection (both here and in 1954), arguing that there is often a fitness peak for the quantitative character, which "is not usually, if ever an extreme value, and is often close to the mean" (p. 481, Haldane 1954). However, when calculating the substitution load, Haldane must switch from considering a quantitative trait subject to stabilizing selection to fitness, which is subject to directional selection. As Ewens (1970) showed, Haldane's calculations are not consistent with using a reference genotype that is the best in the current population but rather the best genotype possible given the alleles currently circulating, which typically does not exist.

We now clarify that Haldane's calculation will be inaccurate in cases where the reference genotype is not in the population. There are many possible genetic models which could be considered, and in some (e.g. when beneficial mutations are rare) it might be reasonable to expect the optimal genotype to also exist. We also now borrow the language here to point out in the text that Haldane used infant birth weight as his example (in Haldane 1954 and 1957), suggesting that he was considering traits under stabilizing selection where this assumption is indeed reasonable.

Line 145 - This isn't clear without reading Desai and Fisher. When stating that the lead is "equal to the fitness difference between the mean individual and the best genotype present", it would be natural to use the definition on line 134 of fitness (W as "absolute survival"), but Desai and Fisher were working with Malthusian fitnesses (where $m_{\text{MEAN}} - m_{\text{BEST}} = \log(W_{\text{MEAN}}/W_{\text{BEST}})$). Also, shouldn't it be the "fitness difference between the best genotype present and the mean" (lead is positive, right?).

We have completely rewritten this section to now introduce the concept of load entirely in one place and explicitly define lead and lag load. We also reworded to ensure positive lead. We now introduce lead in a conceptual manner that sidesteps the annoying accounting difference between per-generation and Malthusian fitness, which are not important to the point here. We do however now note the difference as a minor matter of bookkeeping in the Figure 2 legend.

Line 146 - Now the selection intensity seems to have flipped back to referring to the "best genotype present" (not the best genotype possible).

In our revision, selection intensity now clearly always refers to the best genotype present.

Line 149 - Define lag load explicitly here (it's unclear given the back-and-forth about how to interpret Haldane above)

We have rearranged for clarity, creating less back and forth. Lag load is now clearly contrasted with lead, and separated from our walkthrough of Haldane 1957, given that he does not explicitly use the term load.

Line 151 - "is normally defined as Haldane did, with respect to the best genotype that can be constructed from segregating alleles" Again, I don't think that was what Haldane did, but it might be what Haldane's calculations implicitly did (Ewens 1970).

We have removed this sentence, to avoid giving the impression that Haldane intended this formulation. While discussing Haldane's calculations earlier, we now explicitly present where we believe Haldane's calculation includes this implicit assumption.

Line 208 - Haldane's absolute fitness? The selection intensity is relative ($\ln(s_0/s)$), so even if absolute survival rates were discussed, a shift from absolute to relative fitness is immaterial? Clarify line 208.

We have expounded on Maynard Smith's model to draw out the contrast with Haldane's model more clearly. Instead of referring to lag load vs. lead, we refer to the assumption of infinite reproductive excess, which we think makes the same point more clearly.

Line 246 - "Where Haldane compared the mean fitness of the population to the mean fitness of a hypothetical population" - Did he? Where?

Our language was imprecise, we have clarified that Haldane's calculation includes a comparison to an optimal genotype, which is not required for Nei and Felsenstein's calculations.

Line 364 - I couldn't follow this. Where does $k_{\text{mean}} - (1/k_{\text{best}})$ come from? Is this fraction formatted correctly?

We now give the calculation in more detail in the text.

Lines 561-566 - As noted by Crow (1970, Genetic Loads and the Cost of Natural Selection),

load is defined *relative* to a reference fitness in the denominator (otherwise multiplying all fitnesses by some arbitrary constant c changes the load, which it shouldn't). Make the definitions precise (e.g., with formulae along the way) to clarify what you intend to be measuring as Malthusian fitnesses (as in Desai and Fisher for lead) or W 's.

Clarified the glossary, including equations.

December 22, 2024

RE: GENETICS-2024-307724

Prof. Joanna Masel
University of Arizona
Ecology & Evolutionary Biology
1041 E Lowell St
Tucson, Arizona 86721

Dear Dr. Masel:

Congratulations! We are delighted to inform you that your manuscript entitled "Substitution load revisited: a high proportion of deaths can be selective" is acceptable for publication in GENETICS. Many thanks for submitting your research to the journal.

During this final evaluation, I did notice a few minor issues, which can be addressed without further review. You can view these comments at the bottom of this email .

I thank the authors for their attention to previous suggestions made by myself and by the reviewers (passed along to GENETICS). The revision has addressed my remaining concerns. The manuscript now provides an interesting overview of the substitution load and the resulting speed limit placed on the incorporation of beneficial mutations. The paper reviews previous approaches, detailing their differences, and extends the arguments to species with multiple life stages. I am pleased to accept your paper.

To Proceed to Production:

1. Format your article according to GENETICS style, as discussed at <https://academic.oup.com/genetics/pages/general-instructions>, and upload your final files at <https://genetics.msubmit.net>.
2. Your manuscript will be published as-is (unedited-as submitted, reviewed, and accepted) at the GENETICS website as an Advanced Access article and deposited into PubMed shortly after receipt of source files and the completed license to publish. Please notify sourcefiles@thegsajournals.org if you do not wish to publish your article via Advanced Access.
3. We invite you to submit an original color figure related to your paper for consideration as cover art. Please email your submission to the editorial office or upload it with your final files. You can submit a small-sized image for evaluation, and if selected, the final image must be a TIFF file 2513px wide by 3263px high (8.375 by 10.875 inches; resolution of 600ppi). Please avoid graphs and small type.

If you have any questions or encounter any problems while uploading your accepted manuscript files, please email the editorial office at sourcefiles@thegsajournals.org.

Sincerely,

Sarah Otto
Associate Editor
GENETICS

Approved by:
Howard Lipshitz
Editor in Chief
GENETICS

note: Please add jnls.author.support@oup.com and genetics.oup@kwglobal.com (or the domains @oup.com and @kwglobal.com) to your email program's "safe senders" list. You will be contacted by both at various points during the production process.

Review comments (if applicable):

The introduction would flow better if the sentences starting on lines 55, 59, and 71 were deleted and merged into the sentences

on line 69 & 70: "This assumed 10% limit was largely incorporated in subsequent work, without strong evidence. Although Haldane's logic has been challenged on multiple counts (Ewens 1970; Felsenstein 1971; Kern 59 and Hahn 2018; Maynard Smith 1968a), Nei (1971) and Felsenstein (1971) derived a near-identical speed limit to Haldane's, using a model that more explicitly captures the finite nature of reproductive excess." [E.g., this avoids the use of "flaw" to refer to having limited data upon which to estimate the % of selective deaths.]

There are odd "_" at the beginning of equations on lines 247, 249, and 338.

The sentence on line 377 needs refining. "Per-generation fitness is the product of fitness components" is true for viability, but not for fecundity. Maybe define fitness component here as the number of surviving individuals and offspring per individual in the previous life stage? It is also worth noting that early versus late reproduction have unequal effects on fitness in populations that aren't constant in size, not captured by the product.

Line 530 should use p_0 for the initial frequency not $-\ln(p_0)$.

Line 572 - Generation is defined differently in demographic models (not first possible return) and also depends on whether the population is growing or shrinking. See, e.g., Charlesworth's (1994) book on evolution in age-structured populations.

Line 585 - Load should mention the "relative difference" or "proportional difference" not just "difference", so it is clearer that this difference is scaled to some fitness measure.